# Can additional funding improve mental health outcomes? Evidence from a synthetic control analysis of California's millionaire tax

Michael Thom *

Price School of Public Policy, University of Southern California, Los Angeles, California, United States of America

* mdthom@usc.edu

**Data Availability Statement:** All the data used in this study are publicly available. The Center for Disease Control and Prevention's National Vital Statistics System mortality data is held in a public

## Abstract

California is the only one of its peers with a state-wide tax earmarked for mental health programs. The voter-approved levy applies to personal income above $1 million and has generated over $20 billion since 2005. But whether the additional funding improved population mental health remains unknown. This study applies the synthetic control method to the CDC's National Vital Statistics System data to determine how the tax affected suicide deaths in California. Findings show that the state's suicide mortality rate increased more gradually after the tax's implementation than it would have otherwise. By 2019, the cumulative impact was approximately 5,500 avoided deaths. Multiple robustness and sensitivity checks confirm that result. However, the effect did not appear immediately, nor was it present within all demographic groups. Nevertheless, additional revenue was associated with improved mental health in California. Other governments may likewise yield beneficial outcomes.

## Introduction

Population mental health deteriorated over recent decades. Depression prevalence rose in the 1990s and 2000s, especially among adolescents and young adults [1,2]. From 1999 through 2019, the suicide mortality rate in the United States increased by over 30 percent [3]. Suicides now outnumber homicides by more than 2:1, and it is the second leading cause of death among individuals under 34 [4].

The trend does not have a single catalyst. Culprits include rising social isolation, economic instability, and technology use [5–7]. Mental health treatment is expensive for many, some doubt its effectiveness, and stigma makes others reluctant to seek help [8,9]. Public policy also bears responsibility. Deinstitutionalization, which encouraged the closure of state psychiatric hospitals starting in the 1960s, left many who needed inpatient care without viable options, and mental health funding was a common target of state budget-balancing in the 1980s and 1990s [10].

Although declining mental health is a national dilemma, public funding obligations largely remain with state governments. All 50 draw on general tax revenue to underwrite costs, and

repository at https://www.cdc.gov/nchs/nvss/deaths.htm.

**Funding:** The author(s) received no specific funding for this work.

**Competing interests:** The authors have declared that no competing interests exist.

five use earmarked taxes to raise additional resources. Colorado, Illinois, Missouri, and Washington permit municipalities, typically counties, to levy a sales or property tax surcharge to subsidize mental health programs. But since not all municipalities exercise the option, none of those taxes is state-wide.

California's strategy differs. The state's Mental Health Services Act imposes a one percent "millionaire tax"—i.e., a personal state income tax on earnings above $1 million. Voters approved the levy in November 2004 amid growing alarm over homelessness and diminishing mental health funding. Notably, the law includes a provision that bars California from shifting the state's pre-existing funding responsibilities to municipalities. In other words, the state cannot use earmarked tax revenue to subsidize previous commitments. The stipulation is consistent with the law's objective to increase mental health funding, not merely develop a new revenue source. From 2005 through 2019, the tax raised approximately $21.6 billion. In 2019 alone, it generated an all-time high of $2.4 billion, or about $60 per capita (Fig 1).

California distributes the revenue to each county based on a formula that incorporates population and cost of living adjustments. Each one develops a stakeholder-informed plan to spend its allocation within three program areas. Community Services and Support (approximately 72 percent of funding in 2019) finances direct services for adults and minors with serious mental health conditions. It can include psychiatric treatment and therapy, substance abuse treatment, and outreach for at-risk populations. Prevention and Early Intervention (approximately 18 percent of funding in 2019) finances services that prevent mental health conditions from instigating severe outcomes, such as hospitalization, homelessness, and suicide. It may include traditional treatment or support for crisis intervention facilities and programs. Innovation (approximately five percent of funding in 2019) subsidizes new, sometimes experimental treatment approaches, such as web-based chat and triage services. Administrative costs and reimbursements consume the remaining funds.

Whether earmarked mental health tax revenue yields population-level improvement remains an open question. Available studies are few and narrow in scope [11]. For example, patients at five tax-supported clinics in Los Angeles County, California, accessed more mental health services, but providers reported higher stress and lower morale [12]. According to other Los Angeles-based assessments, tax-supported programs improved vulnerable groups' mental health [13] and lowered emergency room visits [14], but the progress may have been transitory [15].

This study is the first to examine whether revenue from California's earmarked mental health tax affected a vital public health indicator: suicide mortality. Self-inflicted deaths are perhaps the most heartrending consequence of mental anguish, but they are exceedingly preventable. Indeed, cognitive behavioral therapy and other interventions reduce suicidal ideation and death [16]. To determine the revenue's impact, this study utilizes the synthetic control method to compare outcomes before and after California voters approved the tax. The results show that the state's suicide mortality rate was lower than it would have been without the tax. However, the effect did not emerge immediately, nor did it materialize within all demographic groups. Nevertheless, the results indicate that additional funding can stem the tide of declining mental health and quite literally save lives.

## Methods

### Empirical strategy: The synthetic control method

This study applies the synthetic control method ("SCM") to California to determine whether revenue from the state's earmarked mental health tax improved population mental health as indicated by the state's suicide mortality rate. California is an exemplary case study. It is the only state with a tax that generates funding for all its counties. Unlike staggered local tax

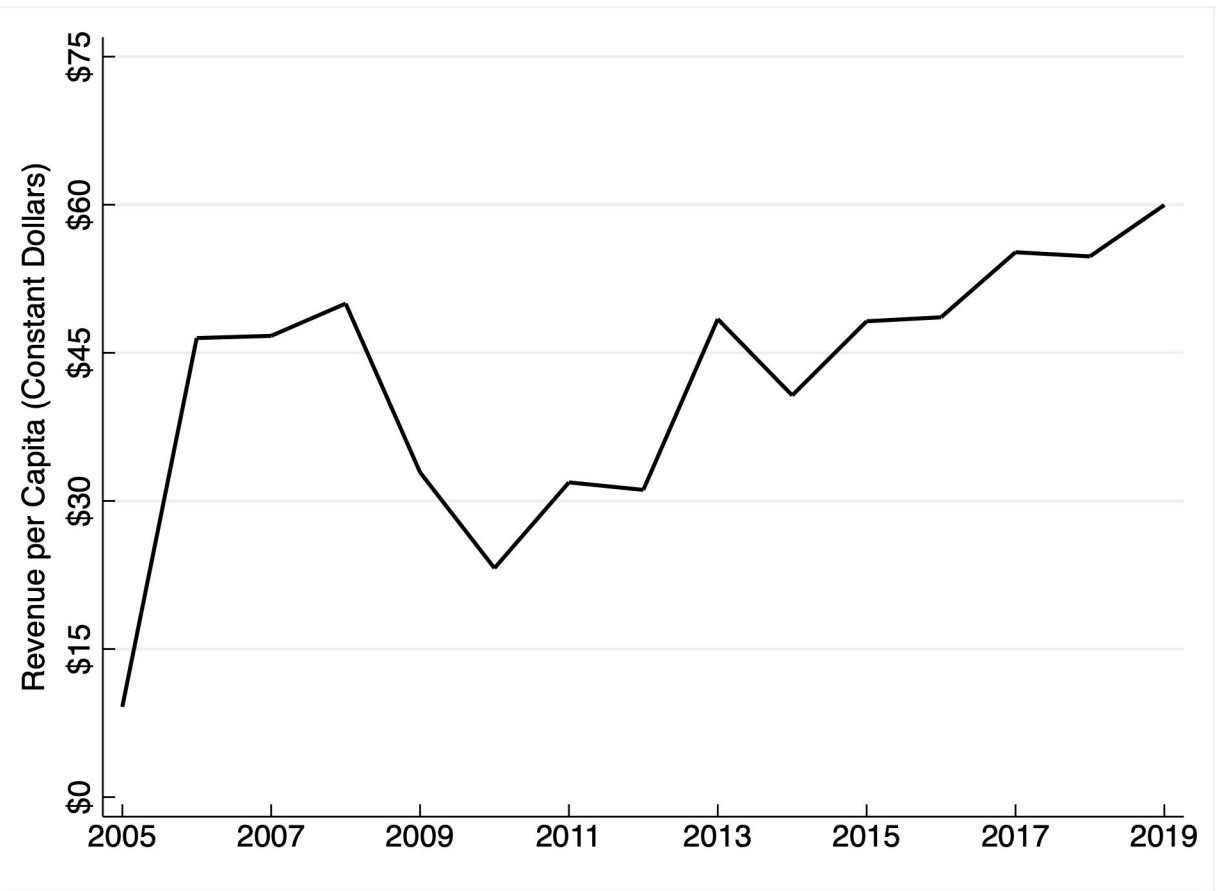

**Fig 1. California mental health services tax revenue per capita.** Author's calculations based on data reported in "Promises Still to Keep: A Decade of the Mental Health Services Act" from California's Little Hoover Commission, annual expenditure reports from the California Health and Human Service Agency, and current population estimates from the United States Census Bureau. Amounts adjusted using the Consumer Price Index.

implementation in other states, California imposed its tax at a single point with no significant ex-post changes, creating well-defined before and after periods.

SCM facilitates causal inference in settings where one unit (here, California) experiences an intervention (here, implementing an earmarked mental health tax) that many other units do not. To synthesize a control, SCM draws on pre-intervention outcomes in California and certain other states, collectively referred to as the donor pool. That synthetic control is a weighted linear combination of pre-intervention donor state outcomes. The method assigns a weight to each donor through a "best fit" optimization procedure that minimizes the pre-intervention discrepancy between California and the control. SCM uses donor state outcomes to estimate the synthetic control's post-intervention trajectory. Differences between California and the synthetic control are construed as an intervention-driven treatment effect. Abadie and Gardeazabal [17] developed the method and applied it to terrorism's economic impact, while others have used it to evaluate a variety of public health policies, including expanded contraceptive availability [18] and soda and tobacco taxes [19,20].

SCM's causal inference capacity rests on constructing a valid counterfactual: the synthetic control. That, in turn, depends on addressing the four empirical issues considered in the following sections.

## Outcome measure and data

Longitudinal, state-level mental health data in the United States is remarkably lacking. The National Survey on Drug Use and Health did not include multiple items on mental health annually until 2009, long after California implemented its tax [21]. The CDC's Behavioral Risk Factor Surveillance System includes mental health items, but not consistently and not necessarily questions of evaluative use [22]. For instance, the survey's 2019 questionnaire included a cumulative question absent in some prior years that asked the respondent if they had *ever* been told they had depression. Another question in 2019 and in preceding years instructed the respondent to estimate the number of days in the prior *month* that their mental health, "which includes stress, depression, and problems with emotions," was "not good." These items mix professionally diagnosed conditions—which may indicate waning outcomes, easier treatment access, or both—with self-diagnosed conditions.

Suicide mortality data is more reliable and available. The CDC's National Vital Statistics System reports each state's annual suicide mortality rate, defined as the number of intentional, self-inflicted deaths per 100,000 persons, from 1999 through 2019 for the general population and demographic groups [23]. The data align with the International Classification of Diseases, 10th Revision, and comprise all tracked methods of intentional self-harm. They exclude accidental self-harm, including opioid and other drug overdoses. The CDC culls the information from death certificates and combines it with Census Bureau estimates to calculate each state's annual age-adjusted mortality rate. Pre-1999 data, while available, are not equivalent. It relies on a somewhat different methodology, does not include racial groups beyond "white," "black," and "other," and comes from a pre-internet social and economic environment.

Data on a less tragic outcome may be preferable but do not exist in adequate quantities. Still, suicide mortality may be the ultimate test of an earmarked mental health tax's effectiveness. Many public health experts regard suicide deaths as preventable. If tax-supported programs in California successfully alter the mental health treatment landscape by expanding access and focusing on early intervention, fewer individuals should choose to take their own life. Moreover, suicide mortality is an impartial indicator unaffected by individuals' perceptions of their psychological status or changing levels of acceptance of having a mental health condition. Those factors threaten survey data validity.

## Pre- and post-intervention periods

California voters approved the state's tax in 2004, but revenue collection did not commence until 2005. Therefore, 2005 is California's intervention year, its pre-intervention period is 1999–2004, and its post-intervention period is 2005–2019. The six-year pre-intervention period is comparable to other synthetic control studies [24–26]. Outcome measure limitations preclude a longer timeframe, but modified assumptions about intervention timing that lengthen the pre-intervention period to seven and eight years do not affect the results (see the Robustness checks section below).

## Predictor variables

The models include the suicide mortality rate lagged by one year and averaged over the entire pre-intervention period. There is no consensus on whether synthetic control models should incorporate predictors other than the lagged outcome variable; some studies have none, while others have over 40 [27]. In analyses like this study that draws on several pre-intervention periods, predictors beyond the lagged outcome are of little benefit and may bias the results [28,29] partly because the lagged outcome "absorbs" their effect and unobserved factors [19]. Diagnostic testing confirmed that estimation with other predictor sets, including the unemployment

rate, per capita income, and ideology, substantially worsened model fit. Accordingly, the models exclude additional predictors.

## The donor pool

At a minimum, the donor pool used to create the synthetic control cannot include the states with a policy like California's: Colorado, Illinois, Missouri, and Washington. There are no other exclusions on policy grounds because no other state implemented an earmarked mental health tax or an alternative policy with an ostensibly similar effect (e.g., a significant and sustained funding increase). However, the donor pool should not include all remaining states, which could "overfit" the synthetic control and lead to interpolation bias [30].

Instead, it is prudent to restrict the donor pool to states like California—i.e., that have comparable outcomes, some slightly above and others slightly below [31]. Because California's general population suicide mortality rate is consistently among the lowest in the United States, relatively few states fare worse. The donor pool includes all nine states that most often had rates close to but less than California during each pre-intervention year: Connecticut, Hawaii, Illinois, Maryland, Massachusetts, Minnesota, New Jersey, New York, and Rhode Island. In the interest of symmetry, the donor pool includes nine states that most often had rates close to but higher than California during each pre-intervention year: Delaware, Georgia, Iowa, Michigan, New Hampshire, Ohio, Pennsylvania, Texas, and Virginia. S1 Table lists each donor state's weight. Note: "most often" refers to the frequency with which the donor states' mortality rates were close to California's. Each donor state had a similar mortality rate in at least four out of six pre-intervention years. This ensures that the donor pool excludes outliers with only occasional similarity to California.

## Results

### General population findings

Fig 2 shows the age-adjusted suicide mortality rate, measured as self-inflicted deaths per 100,000 persons, for California's general population and the synthetic control. The root mean squared prediction error ("RMSPE") is 0.5904, a goodness-of-fit statistic that indicates the control generally tracks California during the pre-intervention period. The sole anomaly in 2001 results from temporarily lower mortality within specific age groups (see Fig 4). The trendlines remain adjacent shortly following tax implementation but begin to deviate around 2010. California's actual mortality rate increased after that point, but more gradually than the synthetic control. The divergence suggests that additional mental health funding reduced suicide mortality in the general population below the rate that would have occurred in its absence.

Visual comparison alone cannot establish whether the difference—i.e., the treatment effect—was statistically significant. Placebo tests aid that determination by applying SCM to each donor state and comparing donor-control differences, which are random, to the California-control difference. A California-control difference larger than most donor-control differences implies a nonrandom treatment effect; otherwise, what appears as a treatment effect is most likely due to chance [32]. Analysts can visually assess the differences to infer significance, but a probability test that calculates pseudo $p$-values facilitates a more objective appraisal [26,33].

To that end, Table 1 lists the annual treatment effect, measured as the reduction in suicide deaths per 100,000 persons, and its significance from 2005–2019. Because the effect was significant from 2012 forward, although only weakly in 2013 and 2016, reduced mortality in those years was unlikely due to chance and was likely driven instead by additional revenue for mental health programs.

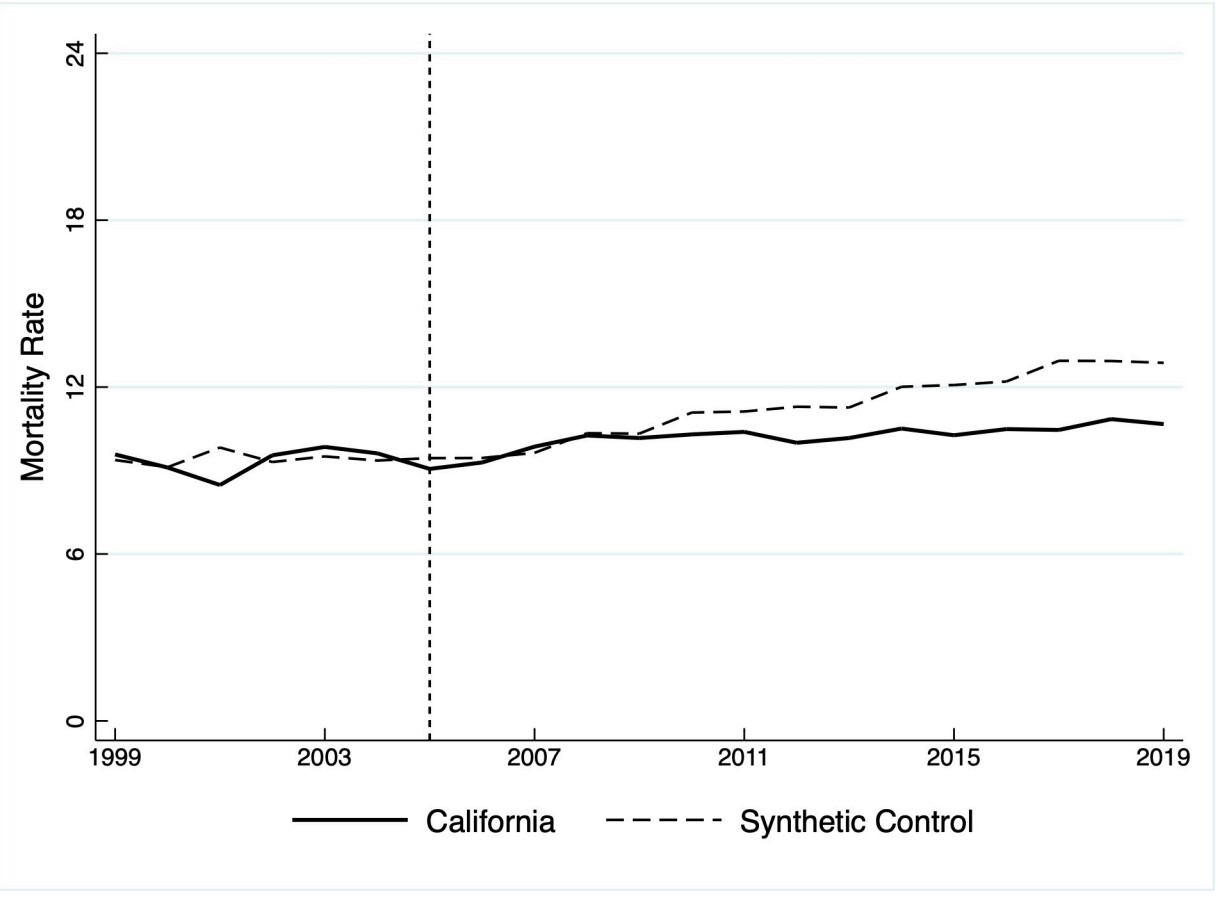

**Fig 2. Age-adjusted suicide mortality rate for California's general population, 1999–2019.**

A back-of-the-envelope calculation translates the tax's effect into real-world impact. In 2019, the effect was a reduction of 2.20 deaths per 100,000 persons. That is the equivalent of approximately 869 suicide deaths avoided that year (the treatment effect multiplied by the state's population of 39.5 million). Using the same method, the cumulative effect since 2012 is about 5,536 deaths avoided. S2 Table reports annual estimates.

## Demographic group findings

Analysis of demographic group-specific data sheds light on whether and how the treatment effect varied among them. The synthetic control approach is identical to that described above, but the donor pools differ. That modification is necessary because the states with suicide mortality similar to California's general population are not the same as those with similar group-specific mortality. Each analysis thus requires a unique donor pool, although there are common donors. S3 and S4 Tables list each analyses' donor states and their respective weights.

Fig 3 shows the age-adjusted suicide mortality rate for California and its synthetic control for six groups: males and females in the top row; whites and blacks in the middle row; and Hispanics and Asians in the bottom row. Table 2 lists the annual treatment effect, statistical significance based on placebo tests, and RMSPE for each analysis.

The impact by sex was more prominent than by race. While it was significant more often for males (eight consecutive years starting in 2012) than females (only four years, but four of

**Table 1. Earmarked mental health tax effect on suicide mortality among California's general population, 2005–2019.**

| Year | Effect | |
|------|-------:|---|
| 2005 | -0.39 | |
| 2006 | -0.16 | |
| 2007 | 0.22 | |
| 2008 | -0.08 | |
| 2009 | -0.16 | |
| 2010 | -0.79 | |
| 2011 | -0.73 | |
| 2012 | -1.30 | *** |
| 2013 | -1.10 | * |
| 2014 | -1.50 | *** |
| 2015 | -1.80 | ** |
| 2016 | -1.71 | * |
| 2017 | -2.48 | *** |
| 2018 | -2.09 | *** |
| 2019 | -2.20 | ** |

The effect is the reduction in mortality rate (deaths per 100,000 persons)

* $p \leq 0.10$

** $p \leq 0.05$

*** $p \leq 0.01$.

the five most recent), the effect on females was proportionally greater. To wit: the reduction in female suicide mortality in California in 2019 was 29 percent (the difference between the actual rate, 4.53, and the synthetic control rate, 6.34). For males, the reduction was only 17 percent (the difference between the actual rate, 17.10, and the synthetic control rate, 20.62). If that continues, it will amplify the already substantial male-female suicide mortality gap.

Effects by race were relatively scarce. There was no reduction in suicide mortality among blacks or Hispanics. The effect on Asians was likewise non-significant but trended in that direction; it rose from -0.06 in 2015 ($p = 0.86$) to -0.93 in 2019 ($p = 0.42$). While the effect on whites appears larger in Fig 2, it was significant only three times in 15 years and never after 2014.

The tax's impact by age was more pronounced than by sex and race. Fig 4 displays the mortality rate for six age groups, and Table 3 lists the annual treatment effect, statistical significance based on placebo tests, and RMSPE for each analysis. The categories stem from the CDC's data, and because they are age-specific, the rates are "crude" rather than age-adjusted. A noticeable effect occurred among those 55–64, a group for whom mortality increased after the tax's implementation but later fell. The most consistent effect happened among those over 65, for whom the reduction increased over time. The tax had less consistent effects on those 45–54 and no effect among those 35–44. With a few exceptions, there is no evidence that the tax affected mortality among those 15–24 and 25–34, the groups for whom suicide became the second-leading cause of death over the same period.

A dearth of more granular data precludes a more in-depth demographic assessment. Simply put, increasingly less information is available on more narrowly-defined groups. The CDC's most available state-level mortality data is for the general population. Missing and unreliable data are more common in more states and years for the black, Hispanic, and especially the Asian population, but sufficient data exists on enough states to compose a donor pool and

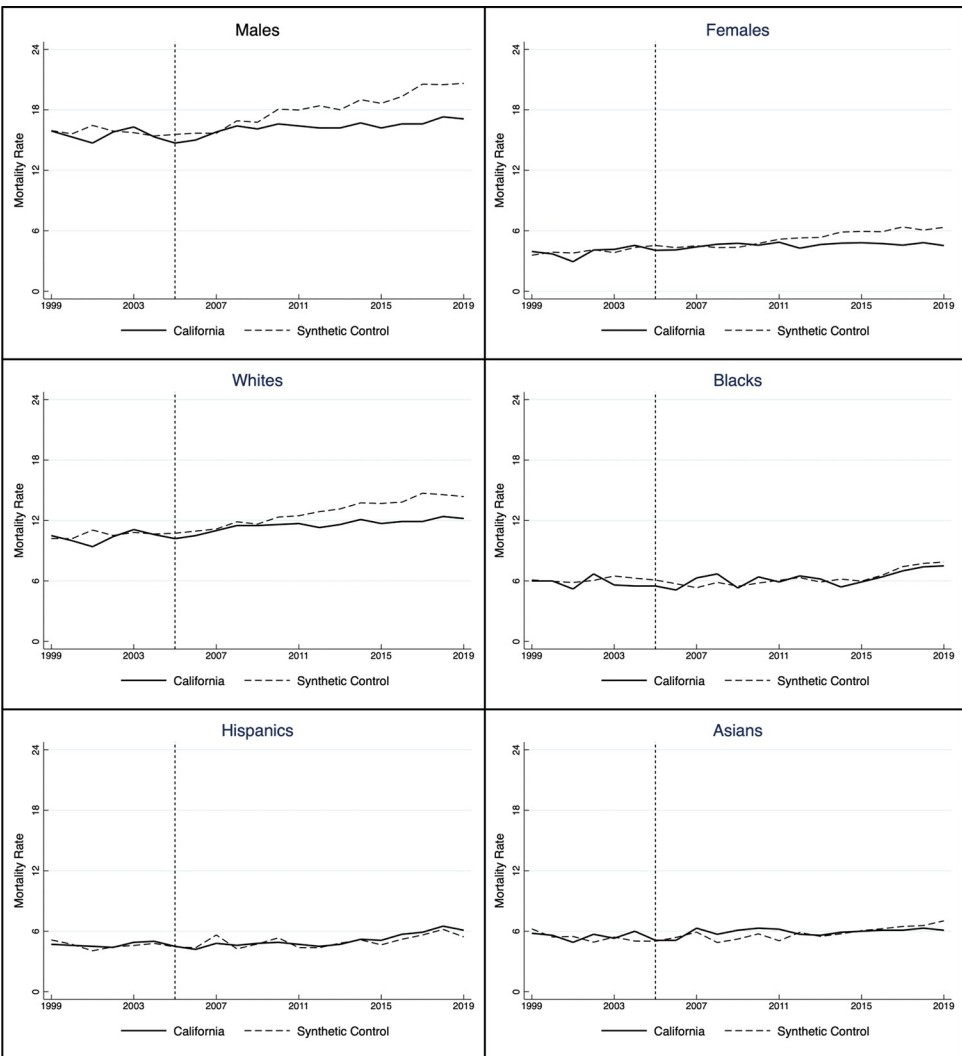

**Fig 3. Age-adjusted suicide mortality rate by sex and race in California, 1999–2019.**

synthesize a control for California. That becomes near impossible with more specific groups (e.g., black males, Hispanic females, or persons under 14 years old). The consolation is this: it is often because there are so few suicides in those groups in many states during some years that the CDC considers the data unreliable.

## Robustness checks

Five robustness checks support the above findings' veracity. Full results for most, which collectively entail over 200 additional synthetic control analyses, are omitted to conserve space. Most of the following discussion centers on general population results, but the robustness checks confirm findings for each demographic group.

First, advancing the intervention date does not substantively alter the treatment effects. S5 Table reports the results for two revised scenarios: an intervention date of 2006, the first full year of substantial tax revenue, and 2007, which allows for a one-year lag between that revenue and a quantifiable effect. Those revisions increase the pre-intervention period to seven and

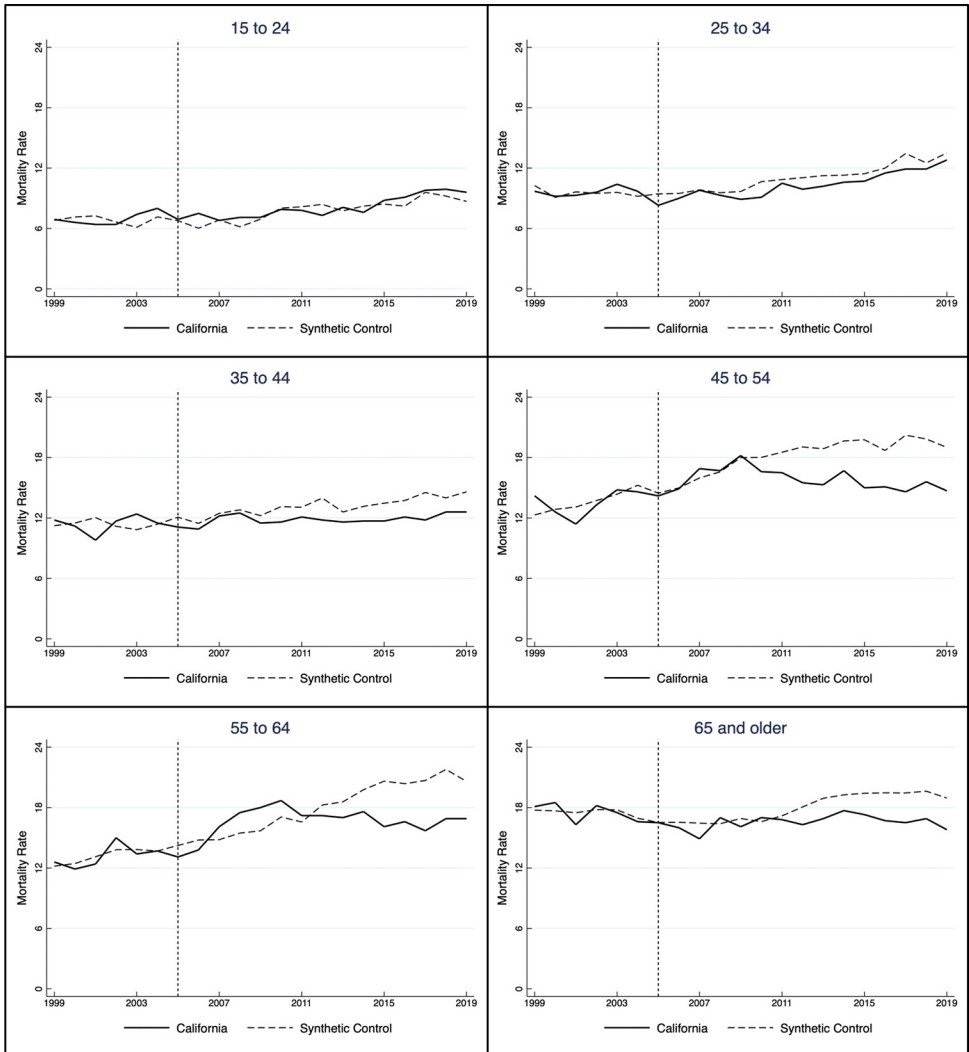

**Fig 4. Suicide mortality rate by age group in California, 1999–2019.**

eight years, respectively, but with little empirical consequence. They correspond to cumulative estimated avoided suicide deaths of 5,376 (assuming an intervention date of 2006) and 5,423 (assuming an intervention date of 2007). Both estimates are within three percent of the number reported above, which derived from a six-year intervention period.

Second, the results are robust to using a broader definition of mortality that combines deaths from intentional self-harm with accidental deaths resulting from substance abuse (e.g., alcohol and drug overdoses, including opioids, cocaine, and sedatives). The mortality rates are necessarily higher, but the pattern repeats: post-intervention, California's rate falls slightly and then rises to a lesser degree than the synthetic control. That finding suggests that additional funding for mental health programs, some of which subsidized substance abuse treatment initiatives, reduced the number of deaths relative to what would have occurred in the funding's absence.

Third, no single donor state biases the findings. Ruling out the prospect of an influential donor(s) entails an iterative "leave one out" tactic suggested by Abadie [34] that systematically repeats each analysis without one of the donor pool states to ensure that no single state biased

**Table 2. Earmarked mental health tax effect on suicide mortality in California by sex and race, 2005–2019.**

| Year | Males | | Females | | Whites | | Blacks | Hispanics | Asians |
|---|---|---|---|---|---|---|---|---|---|
| 2005 | -0.86 | | -0.48 | | -0.54 | | -0.59 | 0.05 | 0.09 |
| 2006 | -0.68 | | -0.23 | | -0.44 | | -0.63 | -0.17 | -0.28 |
| 2007 | 0.13 | | -0.11 | | -0.16 | | 0.97 | -0.83 | 0.39 |
| 2008 | -0.51 | | 0.33 | | -0.37 | | 0.84 | 0.35 | 0.82 |
| 2009 | -0.66 | | 0.41 | | -0.14 | | -0.18 | 0.07 | 0.87 |
| 2010 | -1.46 | | -0.17 | | -0.74 | | 0.61 | -0.45 | 0.56 |
| 2011 | -1.58 | | -0.30 | | -0.77 | | -0.17 | 0.32 | 1.14 |
| 2012 | -2.19 | *** | -1.02 | | -1.58 | ** | 0.16 | -0.13 | -0.19 |
| 2013 | -1.80 | ** | -0.71 | | -1.55 | * | 0.31 | -0.12 | 0.11 |
| 2014 | -2.30 | ** | -1.10 | | -1.65 | ** | -0.78 | 0.05 | 0.16 |
| 2015 | -2.44 | ** | -1.12 | ** | -2.00 | | -0.09 | 0.45 | -0.06 |
| 2016 | -2.72 | *** | -1.17 | | -1.92 | | -0.17 | 0.49 | -0.15 |
| 2017 | -3.94 | *** | -1.81 | *** | -2.81 | | -0.42 | 0.27 | -0.37 |
| 2018 | -3.19 | * | -1.25 | * | -2.17 | | -0.34 | 0.31 | -0.26 |
| 2019 | -3.52 | *** | -1.81 | *** | -2.18 | | -0.39 | 0.65 | -0.93 |
| RMSPE | 0.7566 | | 0.4147 | | 0.7012 | | 0.6186 | 0.3036 | 0.5959 |

The effect is the reduction in mortality rate (deaths per 100,000 persons)

* $p \le 0.10$

** $p \le 0.05$

*** $p \le 0.01$.

**Table 3. Earmarked mental health tax effect on suicide mortality in California by age group, 2005–2019.**

| Year | 15–24 | 25–34 | | 35–44 | 45–54 | | 55–64 | | 65+ | |
|---|---|---|---|---|---|---|---|---|---|---|
| 2005 | 0.13 | -1.14 | | -0.98 | -0.27 | | -1.13 | | -0.02 | |
| 2006 | 1.48 | -0.49 | | -0.57 | -0.09 | | -0.98 | | -0.53 | |
| 2007 | -0.47 | -0.04 | | -0.26 | 0.92 | | 1.29 | | -1.55 | |
| 2008 | 0.93 | -0.25 | | -0.31 | 0.12 | | 2.04 | | 0.60 | |
| 2009 | 0.18 | -0.78 | | -0.74 | 0.20 | | 2.31 | | -0.81 | |
| 2010 | -0.13 | -1.55 | * | -1.53 | -1.42 | | 1.63 | | 0.40 | |
| 2011 | -0.36 | -0.37 | | -0.97 | -2.03 | | 0.63 | | -0.38 | |
| 2012 | -1.10 | -1.14 | *** | -2.18 | -3.56 | | -1.05 | | -1.76 | *** |
| 2013 | 0.35 | -1.04 | * | -1.00 | -3.58 | | -1.57 | | -2.03 | *** |
| 2014 | -0.63 | -0.69 | | -1.45 | -2.95 | | -2.18 | | -1.57 | * |
| 2015 | 0.38 | -0.73 | | -1.78 | -4.77 | ** | -4.53 | *** | -2.13 | *** |
| 2016 | 0.87 | -0.49 | | -1.63 | -3.61 | | -3.77 | *** | -2.76 | *** |
| 2017 | 0.21 | -1.54 | * | -2.73 | -5.63 | *** | -5.00 | *** | -2.95 | *** |
| 2018 | 0.67 | -0.61 | | -1.40 | -4.25 | | -4.88 | ** | -2.73 | *** |
| 2019 | 0.91 | -0.70 | | -2.00 | -4.31 | *** | -3.71 | ** | -3.14 | *** |
| RMSPE | 0.7606 | 0.4733 | | 1.1666 | 1.1059 | | 0.6539 | | 0.6581 | |

The treatment effect is the reduction in mortality rate, measured as deaths per 100,000 persons

* $p \le 0.10$

** $p \le 0.05$

*** $p \le 0.01$.

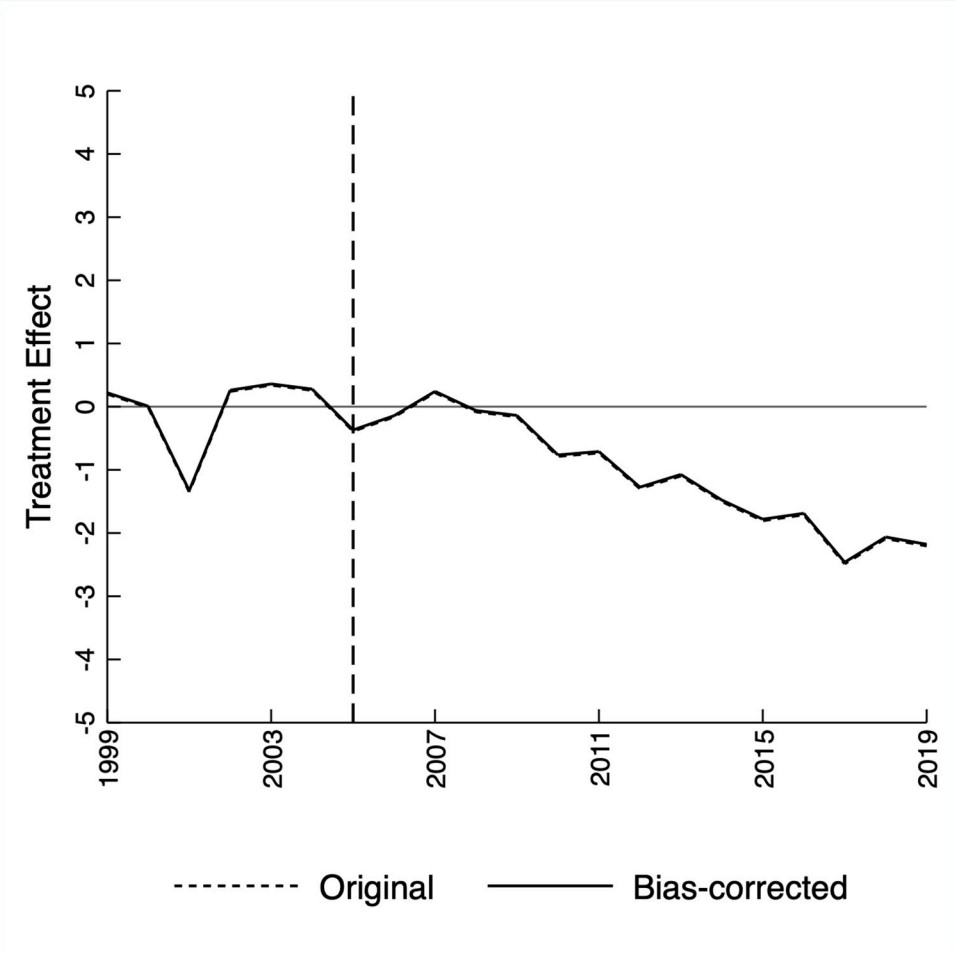

**Fig 5. Original and bias-corrected treatment effects.**

the synthetic control or treatment effects. Omitted donor bias is also not an issue. Re-estimating each analysis to include the next most comparable donors—i.e., adding one extra state with a higher mortality rate and one with a lower rate, where possible—does not yield treatment effects that vary significantly from those reported above.

Fourth, there is no meaningful bias due to imprecise predictor variable matching between California and donor states during the pre-treatment period. Various regression-based techniques exist to estimate and correct such bias, a known issue in SCM [35]. Fig 5 illustrates the original and bias-corrected treatment effects for California's general population using the OLS-based method devised by Wiltshire [36]. The OLS-based method is preferable to those that use ridge, lasso, or elastic net regression because the study's data is straightforward: there is only one predictor variable. The nearly-collinear lines suggest that bias due to imprecise matching does not alter the abovementioned findings. For example, the bias-corrected treatment effect in 2019 was -2.18, which compares favorably to the non-corrected treatment effect reported in Table 1, -2.20.

Fifth, the findings are robust to estimation with an interrupted time-series analysis ("ITSA"). Like SCM, ITSA breaks data into pre- and post-intervention periods and uses the former to estimate an intervention's effect. ITSA is suited for contexts with sequentially

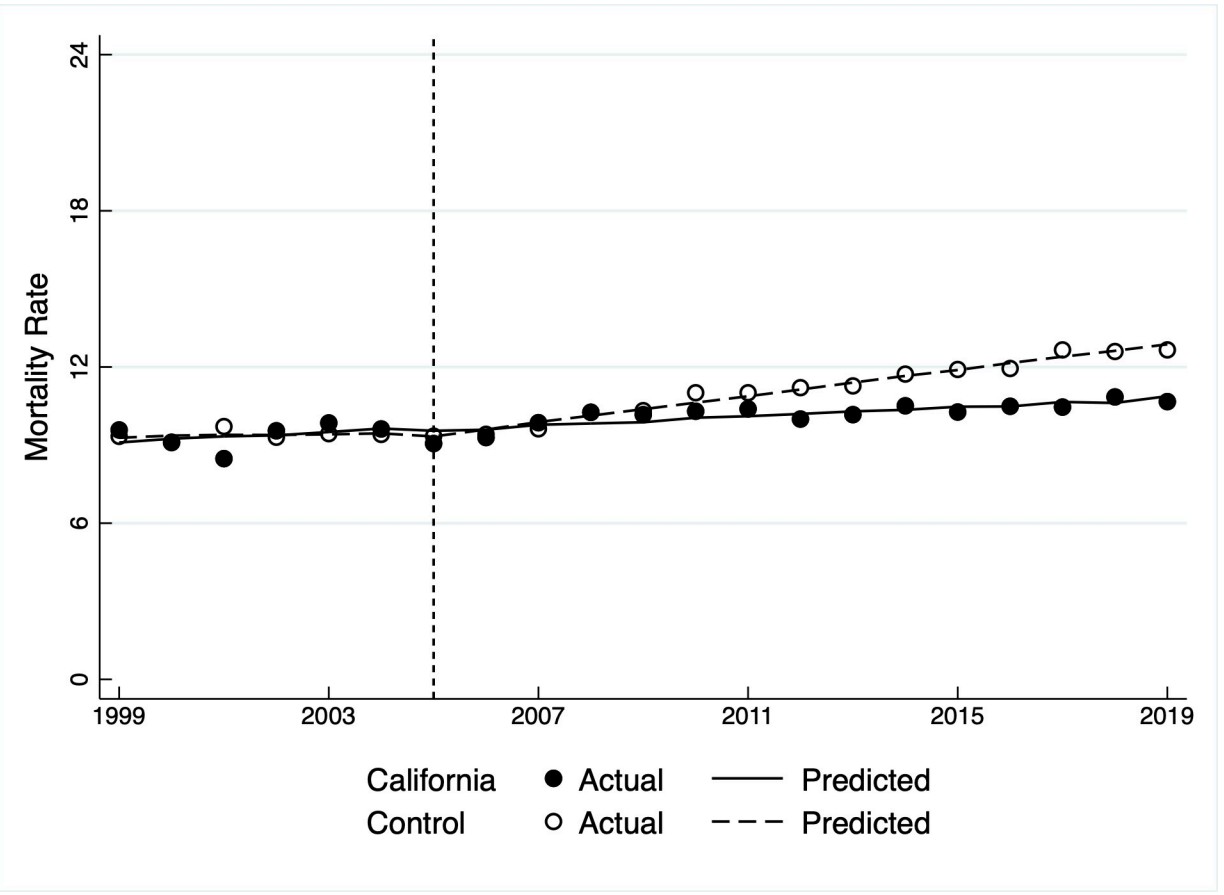

**Fig 6. Actual and predicted suicide mortality rates for California's general population, 1999–2019.**

observed, variable outcome data measured before and after an intervention, making it especially useful for public health evaluations [37,38]. While multiple ITSA iterations exist, this study utilizes Linden's generalized least squares model [39]. It allows a comparison between suicide mortality in California and similar control states. As a robustness check, ITSA is superior to a difference-in-difference model because pre-intervention outcomes in California and control states did not have parallel trends, a violation of the latter method's assumptions that would bias its results. California's pre-intervention period, intervention date, and post-intervention period are unchanged from the SCM approach. The 18 donor states serve as a collective control, and the lagged suicide mortality rate is the sole predictor.

Fig 6 displays the actual and predicted age-adjusted suicide mortality rate for California and the donor state average. Pre-intervention outcomes in California and the control are not significantly different. The post-intervention trendline slope for California is less than the control's, although a gap does not appear visually until around 2010, mirroring the synthetic control findings. The difference, which increases over time, is significant ($p = 0.002$). While the mortality rate in control states continued to rise in the early 2000s—following the nationwide trend that began around the same time—California's increase was comparatively flat. That suggests that, while mortality in California rose, it did so to a lesser extent than it would have in the absence of funding from the state's earmarked mental health tax. Like the SCM results, the ITSA finding is robust to assuming an intervention date of 2006 or 2007. The finding also holds when applied to the broader mortality rate that includes substance abuse deaths.

## Discussion and conclusion

Public policy changes, including deinstitutionalization and diminished funding, have exacerbated recent declines in population mental health. In response, a small number of states permit municipal governments to generate new resources through earmarked mental health taxes. California's state-wide tax on incomes above $1 million is unique. This study assesses whether it affected an essential public health indicator: suicide mortality. It finds that there were fewer suicide deaths after the tax's implementation than there would have been otherwise.

That the effect did not emerge immediately is not surprising. Voters approved the tax in 2004, but it did not take effect until 2005. California did not collect substantial revenue until one year later. Revenue collapsed after 2008 and did not return to its pre-recession level until 2013. As collections recovered and attained stability, the effect on suicide mortality gained statistical and practical significance.

This study's findings comport with prior research linking additional economic resources with improved mental health outcomes at the local level [13,14]. However, in contrast to some others [15], the effect reported herein was durable. The methodology cannot reveal the exact causal path from new funding to reduced suicide mortality, but funding for prevention and early intervention presumably helped more individuals cope with psychological distress before it evolved into a more severe condition. The funding also increased treatment options for those already diagnosed with depression, a frequent conduit to suicide.

Yet this study finds that the effect was inconsistent across demographic groups. Although a reduction in the mortality rate was more frequent among males, the impact was proportionally larger for females. Here, again, the study's methodology cannot reveal the precise cause(s). The etiology may be males' lower likelihood of seeking timely mental health treatment, which arises from stigma and lower perceived need [40,41]. Males that manage to overcome those barriers can fall victim to gender bias. Evidence suggests physicians are less likely to identify male depression than female depression [42]. That is partly due to diagnostic criteria that favor female-typical symptoms [43]. It can also stem from implicit bias: studies show that adults are less likely to recognize depression in males than females [44].

There was little effect among whites and none among California's black, Asian, and Hispanic populations. That may result from male effects canceling out female effects or vice versa. It may also be difficult for public health interventions to reduce mortality within groups with relatively low rates. But those are also groups for whom cultural expectations, along with nativity differences and language barriers, present obstacles to mental health treatment [45,46]. Implicit bias is also a likely factor. As with males, physicians are less likely to diagnose depression in blacks and Hispanics [37], and their treatment may be less tailored than whites' [47,48].

The most visible effects occurred within specific age categories with higher suicide rates: those 55–64 and over 65. The proximate cause is likely a treatment gap. Those 18–25 in 2019 reported a higher level of serious mental illness than those over 50 (8.6 percent versus 2.9 percent) and a higher level of suicidal thoughts (11.8 percent versus 2.4 percent), but a lower level of treatment (56.4 percent versus 74.3 percent) [49]. The root cause for that disparity may be the flexibility that comes with age. Older adults, especially those over 55 or retirees, can easily accommodate inpatient and outpatient care because they are less tethered to fixed employment schedules and domestic responsibilities. That makes it simpler to initiate and sustain treatment. Individuals over 65 may also benefit from Medicare's mental health services, although those services are not without limitation (e.g., only primary care physicians can screen for depression, and many covered treatments have co-pays or deductibles).

This study suggests several policy changes that could improve earmarked mental health taxes and associated programs. They point to a need for responsive programs that address challenges unique to racial and ethnic groups and that also address male-specific concerns. They also suggest a need for improved access for younger adults, especially 15 to 34 year-olds. That may include better integrating mental health care services into high schools and colleges, expanding the availability of paid time off from employment to receive treatment, and deploying mobile app-based diagnostic and treatment tools. Prevention and early intervention programs, which currently receive less than one-fifth of funding in California, could more effectively reduce suicide mortality if counties and the state allocated more resources to them. Moreover, while added funding can yield public health benefits, California's instrument—a tax on personal income over $1 million—is not necessarily ideal. Tax revenue from that income level is notoriously volatile, and a rising number of high-income earners have left California and other high-tax states [50,51]. Both factors threaten funding and policy success.

Finally, additional investigation of the underlying causes of rising suicide rates would also improve policy success in California and beyond. That knowledge is essential for developing programs that reduce intentional self-harm, but the current literature is inconsistent, and the past's conventional wisdom may not comport with the present's reality. For example, although multiple studies suggest a correspondence between unemployment and suicide, others indicate that the link is weak to absent [52,53], is inconsistent [54], or merely coincides with other factors that are the actual cause, such as repeated exposure to bad economic news [55] and stock market volatility [56]. Research on the interaction between suicide and other diseases of despair, especially opioid and other drug overdoses, would also improve mental health program design. Most overdoses are classified as accidental, particularly those involving opioids, yet the recent increase presents no less a formidable challenge to public policymakers. Luckily, with the proper response and adequate funding, progress is attainable.

## Supporting information

**S1 Table. Donor states and weights for synthetic control analysis of California's general population.**
(DOCX)

**S2 Table. Earmarked mental health tax effect on suicide deaths among California's general population, 2012–2019.**
(DOCX)

**S3 Table. Donor pool states and weights for sex and race analyses.**
(DOCX)

**S4 Table. Donor pool states and weights for age group analyses.**
(DOCX)

**S5 Table. Earmarked mental health tax effect on suicide mortality among California's general population using alternative intervention years.**
(DOCX)

## Author Contributions

**Conceptualization:** Michael Thom.

**Data curation:** Michael Thom.

**Formal analysis:** Michael Thom.

**Methodology:** Michael Thom.

**Writing – original draft:** Michael Thom.

**Writing – review & editing:** Michael Thom.

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
