## [Decision Letter · Decision Letter 0]

11 Mar 2022

PONE-D-21-38742Can additional funding improve mental health outcomes? Evidence from a synthetic control analysis of California’s millionaire taxPLOS ONE

Dear Dr. Thom,

Thank you for submitting your manuscript to PLOS ONE. After careful consideration, we feel that it has merit but does not fully meet PLOS ONE’s publication criteria as it currently stands. Therefore, we invite you to submit a revised version of the manuscript that addresses the points raised during the review process.

We look forward to receiving your revised manuscript.

Kind regards,

Mohamed F. Jalloh

Academic Editor

PLOS ONE

Journal Requirements:

2. Please ensure that you include a title page within your main document. You should list all authors and all affiliations as per our author instructions and clearly indicate the corresponding author.

Reviewers' comments:

Reviewer's Responses to Questions

**Comments to the Author**

1. Is the manuscript technically sound, and do the data support the conclusions?

Reviewer #1: Partly

Reviewer #2: Yes

2. Has the statistical analysis been performed appropriately and rigorously? 

Reviewer #1: Yes

Reviewer #2: Yes

3. Have the authors made all data underlying the findings in their manuscript fully available?

Reviewer #1: Yes

Reviewer #2: Yes

4. Is the manuscript presented in an intelligible fashion and written in standard English?

Reviewer #1: Yes

Reviewer #2: Yes

5. Review Comments to the Author

Reviewer #1: 1) General comments:

This paper utilizes suicide mortality data from CDC over a period from 1999-2019 to examine the impact of CA’s mental health tax policy. Its strengths are that it applies synthetic control methods on nationally suicide mortality data to examine the impact of the CA mental health tax policy. The author further explored the policy impact in sub-populations. The disparities among different sub-population provided policy insights. Although suicide mortality is considered as the ultimate outcome of the mental health tax policy and consistently available, there are many other policy and epidemiological factors that would have impact on this outcome. Because of the reliability of the data source, the author uses 5 years pre-intervention data to predict 15 years post intervention outcome without accounting or acknowledging the influence of other noises. Overall, the data and results did not strongly support the assumption that CA’s tax policy has an positive impact on population mental health outcome.

2) Detailed comments

• Data source and outcome measure: The author discussed the availability of national available data as outcome measure of mental health and identified suicide mortality data as the ultimate outcome measure for mental health because it’s consistent, available, and good measure of population mental health. However, in all the models presented, the author only uses the pre-intervention outcome (suicide intervention) as the predictor while there are many other factors related to this outcome (e.g., opioid epidemic in 2010, the great recession in 2008, increased unemployment rate results from the recession, etc.). In fact, the opioid use disorder rates in the selected control states are all very high (majority higher than CA), which might result in greater increase in suicide mortality. And this might be easier to be explained by Fig 2 where the diverge happens around 2010.

In a different thought, is it possible to explore other mental health related outcome data? For example, since 72% of the revenue goes towards individuals with SMI, is it possible to explore the private insurance and state Medicaid data for SMI prevalence and/or inpatient/outpatient visits associated with SMI?

• Citations for NSDUH, BRFSS, and CDC mortality data are needed

• Predictors, Pre- and post- intervention periods: because the reliable suicide mortality data is only available after 1999, 1999-2004 is the pre-period, and 2005-2019 is the post period. The author uses a 5-year pre-intervention data to predict 15 years post intervention trend. The RMSPE statistic is reported and seems a good indicator for goodness-of-fit. There is no arbitrary cut-off value for RMSPE but it seems the 2001 data points for control group is at its peak while CA is at its lowest value. I’m curious about different RMSPE values for different predictor sets, e.g., by adopting the data-driven methods for variable selection described by Abadie (2021).

• Donor Pool: most of the selected states are in the east side, a few in mid-west and none of them is CA’s neighbor state. They are geographically and epidemiologically very different from CA despite the similar suicide mortality rate prior to 2008. Using the OUD rate as an example, the rise of suicide mortality rate post 2010 might just because of the severer impact of the opioid impact on those states.

• Results: the parameter estimation results for ITSA are not provided, but the author provided the p value for the post-intervention slope change. By looking at the figure provided, it looks more like some intervention was done to the control group in 2004 which results in slope change for control group and the policy does not results in any slope change for CA (treatment)? Although parallel trend assumption is not strictly required in ITSA, it is expected that slope and level change for treatment group is larger than the control group. Otherwise, the author may want to explain why there is a slope change in 2004 for control group rather than for CA.

Reviewer #2: In general, this is an interesting paper.

1. Synthetic Control Method is supposed to be a data-driven, transparent, objective evaluation method. The paper imposed several subjective constraints regarding the choice set of the donor pool with some reasoning. However, we still like to see some sensitiveness analysis due to these subjective constraints/exclusion (page 7).

2. There are no details on the modeling approach used and no equations in the paper that would enable us to judge/reduplicate what exactly is being modeled.

3. Suggest applying the permutation test to evaluate the significance and robustness of the estimations.

4. Maybe the paper can intuitively show, for example, how much tax money could deter one suicide life in different age/ethnic/gender groups in California based on your estimation of the synthetic California .

6. PLOS authors have the option to publish the peer review history of their article (what does this mean?). If published, this will include your full peer review and any attached files.

Reviewer #1: No

Reviewer #2: **Yes: **Cheng Yuan

---

## [Author Response · Author response to Decision Letter 0]

23 May 2022

See attached document for my response to the reviewers.

---

## [Decision Letter · Decision Letter 1]

16 Jun 2022

PONE-D-21-38742R1Can additional funding improve mental health outcomes? Evidence from a synthetic control analysis of California’s millionaire taxPLOS ONE

Dear Dr. Thom,

Thank you for submitting your manuscript to PLOS ONE. After careful consideration, we feel that it has merit but does not fully meet PLOS ONE’s publication criteria as it currently stands. Therefore, we invite you to submit a revised version of the manuscript that addresses the points raised during the review process.

We look forward to receiving your revised manuscript.

Kind regards,

Mohamed F. Jalloh, PhD, MPH

Academic Editor

PLOS ONE

Journal Requirements:

Reviewers' comments:

Reviewer's Responses to Questions

**Comments to the Author**

1. If the authors have adequately addressed your comments raised in a previous round of review and you feel that this manuscript is now acceptable for publication, you may indicate that here to bypass the “Comments to the Author” section, enter your conflict of interest statement in the “Confidential to Editor” section, and submit your "Accept" recommendation.

Reviewer #1: All comments have been addressed

Reviewer #2: All comments have been addressed

2. Is the manuscript technically sound, and do the data support the conclusions?

Reviewer #1: Yes

Reviewer #2: Yes

3. Has the statistical analysis been performed appropriately and rigorously? 

Reviewer #1: Yes

Reviewer #2: Yes

4. Have the authors made all data underlying the findings in their manuscript fully available?

Reviewer #1: Yes

Reviewer #2: Yes

5. Is the manuscript presented in an intelligible fashion and written in standard English?

Reviewer #1: Yes

Reviewer #2: Yes

6. Review Comments to the Author

Reviewer #1: I believe my comments are well addressed. I appreciate that the author did additional check on confounding variables such as unemployment rate.

The only suggestions I had is to actually keep the outcome of the original submission (include suicide from overdoses), and make the current revision as a sensitivity check. I see the author's intention of excluding those suicide categories and how minor the changes are. But thinking about where the mental health tax money went, it should address both mental health and SUD and not just depression related suicide. Actually a lot of SUD were treated at MH clinics.

Another thought about suicide and MH data - 911 calls will have records of suicide related cases and you can actually get data about whether referral were made. Those data might also be good national MH data but they are not publicly available as CDC mortality data, the author may need to request. Again it's just a side thought about data availability and potential outcome to investigate.

Reviewer #2: (No Response)

7. PLOS authors have the option to publish the peer review history of their article (what does this mean?). If published, this will include your full peer review and any attached files.

Reviewer #1: No

Reviewer #2: **Yes: **YUAN CHENG

---

## [Editor Report · Decision Letter 2]

23 Jun 2022

Can additional funding improve mental health outcomes? Evidence from a synthetic control analysis of California’s millionaire tax

PONE-D-21-38742R2

Dear Dr. Thom,

We’re pleased to inform you that your manuscript has been judged scientifically suitable for publication and will be formally accepted for publication once it meets all outstanding technical requirements.

Kind regards,

Mohamed F. Jalloh, PhD, MPH

Academic Editor

PLOS ONE
---

## [Editor Report · Acceptance letter]

6 Jul 2022

PONE-D-21-38742R2 

Can additional funding improve mental health outcomes? Evidence from a synthetic control analysis of California’s millionaire tax 

Dear Dr. Thom:

I'm pleased to inform you that your manuscript has been deemed suitable for publication in PLOS ONE. Congratulations! Your manuscript is now with our production department. 

Kind regards, 

on behalf of

Dr. Mohamed F. Jalloh 

Academic Editor

PLOS ONE